# Filling the Void: An Optimized Polymicrobial Interkingdom Biofilm Model for Assessing Novel Antimicrobial Agents in Endodontic Infection

**DOI:** 10.3390/microorganisms8121988

**Published:** 2020-12-14

**Authors:** Sumaya Abusrewil, Jason L. Brown, Christopher D. Delaney, Mark C. Butcher, Ryan Kean, Dalia Gamal, J. Alun Scott, William McLean, Gordon Ramage

**Affiliations:** 1Glasgow Endodontics & Oral Sciences Research Group, Glasgow Dental School, School of Medicine, Dentistry and Nursing, College of Medical, Veterinary and Life Sciences, Glasgow University, Glasgow G2 3JZ, UK; 2224354a@student.gla.ac.uk (S.A.); Jason.brown@glasgow.ac.uk (J.L.B.); c.delaney.1@research.gla.ac.uk (C.D.D.); 2135158B@student.gla.ac.uk (M.C.B.); 2418638G@student.gla.ac.uk (D.G.); james.scott@glasgow.ac.uk (J.A.S.); William.mclean@glasgow.ac.uk (W.M.); 2Department of Biological Sciences, Glasgow Caledonian University, Cowcaddens Road, Glasgow G4 0BA, UK; ryan.kean@gcu.ac.uk

**Keywords:** endodontic, biofilm, interkingdom, chitosan

## Abstract

There is a growing realization that endodontic infections are often polymicrobial, and may contain *Candida* spp. Despite this understanding, the development of new endodontic irrigants and models of pathogenesis remains limited to mono-species biofilm models and is bacterially focused. The purpose of this study was to develop and optimize an interkingdom biofilm model of endodontic infection and use this to test suitable anti-biofilm actives. Biofilms containing *Streptococcus gordonii, Fusobacterium nucleatum*, *Porphyromonas gingivalis*, and *Candida albicans* were established from ontological analysis. Biofilms were optimized in different media and atmospheric conditions, prior to quantification and imaging, and subsequently treated with chlorhexidine, EDTA, and chitosan. These studies demonstrated that either media supplemented with serum were equally optimal for biofilm growth, which were dominated by *S. gordonii*, followed by *C. albicans*. Assessment of antimicrobial activity showed significant effectiveness of each antimicrobial, irrespective of serum. Chitosan was most effective (3 log reduction), and preferentially targeted *C. albicans* in both biofilm treatment and inhibition models. Chitosan was similarly effective at preventing biofilm growth on a dentine substrate. This study has shown that a reproducible and robust complex interkingdom model, which when tested with the antifungal chitosan, supports the notion of *C. albicans* as a key structural component.

## 1. Introduction

Endodontic disease is primarily driven by a microbial insult to the pulpal tissues and establishment of infection within the root canal system including the dentinal tubules. Understanding which species are present in endodontic infections will provide important insights into the management of such diseases. Until recently, the field of endodontic microbiology has been swamped with studies investigating *Enterococcus faecalis* [1]. However, with the advent of next generation sequencing, combined with more careful and accurate sampling procedures, we have developed a greater insight into the polymicrobial nature of the infected root canal [2]. These microbiome studies have illuminated our limited understanding by revealing a great diversity of bacteria that represent many of the common ‘garden variety’ oral species associated with cariogenic and periodontal diseases. To make matters more complex, mycobiome studies and other mycological investigations have revealed that *Candida* spp. play an important, yet underplayed, role in supporting these complex communities [3]. Indeed, the interkingdom and polymicrobial nature of these consortia exist within complex polymer-enclosed biofilms that demonstrate inter-species and inter-kingdom interactions, which may be synergistic or antagonistic in nature [4]. The underlying mechanisms of polymicrobial interactions that lead to inflammatory diseases in the oral cavity remain largely unknown [5], however, it is likely that these play a significant role in the development and progression of endodontic disease. Regardless, these biofilm structures demonstrate inherent tolerance to physical and chemical intervention, thus creating significant challenges for clinical management. Indeed, the substantial role of intraradicular infections was established in the form of biofilms in the root canal system of teeth with apical periodontitis [6]. Biofilms have been observed obstructing the walls of apical ramifications, lateral canals, and anastomoses in untreated and treated apical root canals with apical periodontitis [7].

Model systems are a requisite in supporting the development and management of novel physical and chemotherapeutic approaches including to investigate and modify existing protocols. A vast plethora of polymicrobial oral biofilm models have been developed that are now commonplace in most research laboratories investigating the pathogenicity of different oral diseases [8]. However, to date, a limited number of endodontic models exist that accurately recapitulate the microenvironment of the root canal. Existing endodontic biofilm models have been very limited in scope and complexity. It was highlighted that 86% of existing endodontic models reported in the literature consisted of simple single-species biofilms, with 92% of these studies containing only *E. faecalis*. Conversely, of the remaining models reported, only 11 contained two or more species [1]. An improvement in sequencing techniques and sampling methodologies has allowed researchers to show that the infected root canal is an ideal microenvironment for polymicrobial interactions [2]. Indeed, it has been estimated that 10^3^ to 10^8^ bacteria are generally recovered from the infected root canal comprising of approximately up to 20 species [9]. It is postulated that through the loss of blood circulation in necrotic root tissue and oxygen-consuming traits of the existing microbiota, the infected root canal can consist of aerobic and anaerobic microorganisms in an otherwise oxygen-rich microenvironment [10]. These findings represent the significant issues facing innovations in the endodontic field. Two recent systematic reviews reported that next generation sequencing showed that the key genera in primary and persistent infected root canals were *Prevotella*, *Fusobacterium*, *Porphyromonas*, *Parvimonas*, and *Streptococcus* [11], and the most abundant phyla were *Firmicutes*, *Bacteroidetes*, *Proteobacteria*, *Actinobacteria*, and *Fusobacteria* [12]. Therefore, ideally, when it comes to selecting the microbial composition for an optimized endodontic biofilm model, then typical members of microbiota that are commonly isolated from infected root canals are included [1]. Such engineered biofilm systems with defined laboratory strains provide an opportunity to control the model, opposed to undefined ‘natural’ consortia that are inherently problematic, which has important repercussions in testing and developing new endodontic irrigants. These systems are commonly used by different laboratories pertaining to be highly reproducible, easily manipulated, time and cost effective, and rather simply analyzed and interpreted [8,13], whereas undefined consortia are extremely diverse microbial populations that are imprecise for replicate experiments by researchers [14].

With these issues to consider, we sought to develop and optimize a defined multi-species interkingdom biofilm model comprised from a number of oral-species frequently identified in endodontic infection. This was then used to test and evaluate the effect of established “gold-standard” endodontic irrigants and a novel agent (chitosan) that could be employed in root canal treatments. Here, we report the development of a robust optimized 4-species endodontic model biofilm system that was able to demonstrate the positive benefit of chitosan in the prevention and treatment of these consortia.

## 2. Materials and Methods 

All media and reagents were obtained from Sigma-Aldrich (UK) unless otherwise stated. Where appropriate, all biofilm studies reported in this work was carried out in accordance with the minimum information guidelines specified for biofilm formation in microplates [15]. All experiments were performed at least three times with at least three technical replicates, unless otherwise stated.

### 2.1. Ontological Analysis to Inform Biofilm Model Development

Two separate extensive literature searches were performed using the Web of Science: Core Collection (WOS) and PubMed Advanced Search Builder. First, WOS was used to identify original research articles related to the field of microbiology in endodontic infections from 1990 to June 2020. The key search terms included “endodontic infection” OR “root canal infections” OR “apical lesions” AND “microbiome” OR “mycobiome” (Appendix A). This search strategy identified a total number of 8596 publications, which were then imported into EndNote X8. After the removal of reviews, editorial and book sections, a total number of 7569 original research articles were included. Next, these journal references were grouped in single years and exported into a BibTeX format where each entry was described by a number of article records including author, title, journal, a digital objective identifier (DOI), a unique accession number or document ID. Following this, the BibTeX files were processed by a custom-written script called “PyTag” as described previously by [16]. The PyTag script accepts the BibTeX files in a given folder as input and supports nine ontologies used in the EXTRACT 2.0 system [17]. The ontologies recovered mentions for the following: biological process, cellular components, chemical compounds, disease, environment, gene and proteins, molecular function, tissue, and organism. PyTag then utilized these ontologies on the WOS abstracts using the associated IDs. Following the annotation process of all abstracts, the resulting mean of the identified ‘organisms’ terms was converted to two-dimensional tables. Graphs were then created using GraphPad Prism (version 8). 

Next, we used PubMed to identify original studies that utilized next-generation sequencing to characterize the microbiota associated with infected root canals. The key search terms included “endodontic infection” OR “root canal” AND “microbial communities” OR “16S rRNA” AND “next-generation sequencing” OR "pyrosequencing" (Appendix A). Based on this search strategy, a total of 21 original research articles were identified and selected. The top bacterial genera identified in each of these studies were extracted for further analysis using GraphPad Prism. 

### 2.2. Biofilm Model Development

#### 2.2.1. Microbial Standardization

*Candida albicans* SC5314, *Streptococcus gordonii* ATCC 35105 *Porphyromonas gingivalis* ATCC 33277, and *Fusobacterium nucleatum* ATCC 10953 were used throughout this study. All strains were stored at −80 °C on Microbank^®^ vials (Pro-Lab Diagnostics, Birkenhead, UK) prior to propagation on agar media. *C. albicans* was propagated on Sabouraud’s dextrose agar (SAB), and *S. gordonii* grown on Columbia agar supplemented with 5% horse blood (CBA), whilst the two anaerobic organisms were maintained on Fastidious anaerobic agar (FAA) base plates containing 5% defibrinated horse blood, respectively. All agar bases were supplied by Oxoid, UK. *C. albicans* was grown at 30 °C aerobically for 48 h, *S. gordonii* at 37 °C, 5% CO_2_ for 24–48 h, and *P. gingivalis* and *F. nucleatum* at 37 °C in an anaerobic chamber (Don Whitley Scientific Limited, Bingley, UK) with an atmosphere of 85% N_2_, 10% CO_2_, and 5% H_2_ for 24–48 h. 

For broth cultures, *C. albicans* was grown in 10 mL yeast peptone dextrose (YPD) for 16–18 h at 30 °C in an orbital benchtop shaker at 200 rpm, 20 mM orbital diameter (IKA KS 4000 I control, Staufen, Germany). *S. gordonii* was grown statically in 10 mL trypticase soy broth (TSB) at 37 °C 5% CO_2_ for 16–18 h. *F. nucleatum* and *P. gingivalis* were propagated in 10 mL Schaedlers (SCH) broth (Oxoid, Basingstoke, UK) under anaerobiosis at 37 °C for up to 16–18 h. For anaerobic growth, all media and agar to be used in anaerobic chamber were deoxygenated for 24 h prior to use. Negative control broths (minus inoculum) were included to assess for microbial contamination.

Following growth, microbial cells were then pelleted by centrifugation and washed via resuspension twice in phosphate buffered saline (PBS) prior to standardization to 1 × 10^8^ cells/mL. For *C. albicans*, yeast cells were counted on a hemocytometer (cell count × dilution factor × volume of square = colony forming unit [CFU/mL]). Bacterial cells were standardized using a spectrophotometer at 550 nm, with absorbance values of 0.5 for *S. gordonii* and 0.2 for *F. nucleatum* and *P. gingivalis*, approx. equating to 1 × 10^8^ cells/mL, as determined by diluting pure colonies using the Miles and Misra colony counting technique [18]. Standardized cultures were used for biofilm development as described below. 

#### 2.2.2. Development of Single and Mixed-Species Biofilm in Microtiter Plates

Standardized cultures of *C. albicans* (1 × 10^8^ CFU/mL) were first diluted to 1 × 10^6^ CFU/mL and the bacteria (*S. gordonii*, *P. gingivalis* and *F. nucleatum* at 1 × 10^8^ CFU/mL) to 1 × 10^7^ CFU/mL in culture broth. The broth consisted of 1:1 mixture of Roswell Park Memorial Institute-1640 (RPMI) with either tryptic soy broth (TSB) media or Todd Hewitt Broth (THB) supplemented with 0.01 mg/mL hemin and 2 µg/mL menadione in a similar manner to as previously described [19]. Single and mixed-species biofilms were grown in pre-sterilized polystyrene 24 well flat-bottom plates (Costar^®^, Corning Incorporated, Corning, NY, USA) in RPMI/TSB or RPMI/THB supplemented with and without 10% fetal bovine serum (FBS) for 24 h and 48 h within three different environments: aerobically (atmospheric), 5% CO_2_, and anaerobically (85% N_2_, 10% CO_2_, 5% H_2_) at 37 °C. Wells containing media only were included as controls to assess for contamination. 

### 2.3. Biofilm Characterization

#### 2.3.1. Quantification of Biomass

Following incubation, the spent supernatant was discarded, and biofilms were washed once in PBS to remove any non-adherent cells. For biomass assessment, standard crystal violet (CV) was used. Briefly, biofilms were stained with 0.05% (*w*/*v*) CV solution for 15 min at room temperature. A stock solution of 1% CV (Sigma-Aldrich, Gillingham, UK) was made using ddH2O and diluted to 0.05% for use. Following staining, biofilms were washed to remove all unbound dye, and 100% ethanol used to destain the biofilm. This was mixed well with a pipette five times and 100 µL transferred to a fresh flat bottom microtiter plate for measurement spectrophotometrically using the microtiter plate reader (Sunrise, TECAN, Theale, UK) at a wavelength of 570 nm. 

#### 2.3.2. Visualization 

For visualization microscopically, biofilms were grown on 13 mm Thermanox^TM^ coverslips (Fisher Scientific, Loughborough, UK) placed within 24-well plates. Biofilms grown in RPMI/TSB + 10% FBS or RPMI/THB + 10% FBS in a 5% CO_2_ incubator at 37 °C were visualized at 24 and 48 h. Following growth, biofilms were prepared for scanning electron microscopy (SEM), as described previously [20]. In brief, biofilms were fixed in a solution of 2% paraformaldehyde, 2% gluteraldehyde, 0.15 M sodium cacodylate buffer, and 0.15% (*w*/*v*) Alcian Blue and stored overnight at 4 °C until further processing. Following fixation, biofilms were treated with a 0.15 M sodium cacodylate buffer rinse and stored at 4 °C. Sodium cacodylate buffer was removed, and subsequently stained with uranyl acetate prior to a series of dehydration steps in 30%, 50%, and 70%, and a 100% ethanol, followed by a hexamethyldisilizane wash. Subsequently, all samples were placed in a desiccator to allow for evaporation. Samples were gold/palladium sputter coated and then mounted. Digital images were acquired using Jeol JSM-IT100 InTouch™ scanning electron microscope and representative images taken at ×1000 and ×3500, respectively.

#### 2.3.3. Quantitative Analysis of Biofilm Composition

Real-time quantitative PCR (qPCR) was performed to quantify the relative composition of the biofilms. Biofilms were grown in 6-well microtiter plates (Corning, Corning, NY, USA) as described above. These microtiter plates were used to ensure sufficient quantities of DNA was extracted from biofilms to permit accurate qPCR analyses. Following maturation, spent biofilm media was removed by pipetting and biofilms washed with sterile PBS. Biofilms were detached using a cell scraper and transferred to 1.5 mL Eppendorf tubes (Greiner Bio-one, Kremsmünster, Austria). DNA was then extracted from the samples using the QIAamp DNeasy Mini Kit (Qiagen, Manchester, UK), according to the manufacturer’s instructions. Following on from this, compositional analyses were enumerated using quantitative PCR (qPCR). In brief, 1 μL of extracted DNA was added to a mastermix containing 10 μL SYBR^®^ GreenER™, 7 μL UV-treated RNase-free water, and 1 μL of 10 μM forward/reverse primers for each microbial species. The primers used were previously published in [21], and are listed in Table 1. For the reaction, the thermal profiles used were as follows: holding stage at 50 °C for 2 min, denaturation stage at 95 °C for 2 min, and then 40 cycles of 95 °C for 10 s, and 60 °C for 30 s using the StepOnePlus Real-Time PCR system and StepOnePlus software version 2.3 (ThermoFisher, Paisley, UK) for data compilation. Samples were quantified by calculating the colony forming equivalent (CFE) based upon an established standard curve of microbial colony forming units ranging from 1 × 10^3^ to 10^8^ CFU/mL, of which, DNA was extracted as above, and each dilution ran in the qPCR. All samples were run in duplicate in the qPCR, with negative control samples containing water, primers, and mastermix only used to assess for DNA contamination.

### 2.4. In Vitro Biofilm Susceptibility Testing

The optimized model system was used to assess the capacity of different endodontically relevant antimicrobials to inhibit biofilm formation. For all biofilm testing, appropriate guidelines were followed as per Clinical and Laboratory Standards Institute (CLSI) methodologies [22]. The antimicrobials tested and their stock concentrations were as follows: 20% chlorhexidine (*v*/*v*) (CHX (Sigma-Aldrich, UK)), 17% (460 mM) ethylenediaminetetraacetic acid (*w*/*v*) (EDTA (Sigma-Aldrich, UK)), and medium molecular weight chitosan (Sigma-Aldrich, UK) prepared to 1400 μg/mL. For the conventional treatments, CHX and EDTA were diluted to 0.2% and 230 mM, respectively. For chitosan testing, it was prepared as described previously [23]. Briefly, 1400 μg/mL chitosan was solubilized in 2% acetic acid under constant magnetic stirring for 24 h at room temperature. The stock solution was then diluted down to 0.7 mg/mL in the appropriate media prior to biofilm treatment. 

For biofilm testing, the effect of each active was assessed against multispecies biofilms grown in optimal conditions in a 5% CO_2_ incubator at 37 °C. Two regimes were tested, the “treatment” group and “prevention” group. In the “treatment” group, biofilms were treated for 24 h with the therapies at concentrations described above, prior to metabolic assessment. Conversely, to more effectively model endodontic protocols, the “prevention” group contained biofilms that were first mechanically disrupted, in order to assess the ability of the test antimicrobial agents to impede biofilm regrowth. Briefly, after biofilm maturation (24 h), the spent biofilm media was discarded, and biofilms washed with PBS. The biofilms were then disrupted with a cell scraper into 1 mL of RPMI/THB ± FBS media and diluted to 1:5 in fresh RPMI/THB ± FBS media. This new inoculum, representative of the 4-species biofilm consortia, was then inoculated into another 24-well tissue culture plate, prior to treatment with each antimicrobial test agent adjusted to twice the desired concentrations described above. This was to obtain final concentrations of chlorhexidine (0.2%), chitosan (0.7 mg/mL), and EDTA (230 mM). Plates were then incubated for an additional 24 h in 5% CO_2_ at 37 °C to allow biofilm growth. Positive control biofilms were run in parallel for this experiment. These controls were disrupted as above, then replaced with fresh RPMI/THB ± FBS media excluding treatment. Following incubation and washing twice with PBS, the metabolic activity was measured by an Alamar Blue metabolic assay. For this, a 1:10 dilution of Alamar Blue™ (Thermo Fisher, UK) was prepared in RPMI/THB ± FBS media. Biofilms were incubated for 1 h in the dark at 37 °C. For all assays, appropriate negative controls minus inoculum were included to assess for media contamination. The resultant fluorescence color changes were quantified at 544/590 (excitation/emission) using the microtiter plate reader (FluoStar Omega, BMG Labtech, Aylesbury, UK). The composition of the regrown biofilms was also assessed using qPCR, as described above.

### 2.5. In Vitro Biofilm Susceptibility Testing on Bovine Dentine Discs

The effect of the antimicrobial ability of chitosan was assessed against mixed-species biofilms regrowth on a biological substrate “bovine dentine discs” placed in a 24-well plate. Bovine dentine discs (Modus Laboratories, Reading, UK) were used in this study. The specifications for these discs were as follows: round cross section, 6–8 mm in diameter, 1.5–2 mm in thickness, polished (P2500) on one side, and perpendicular dentinal tubules orientation. The dentine discs were autoclaved before use (121 °C for 15 min). The composition of the regrown biofilms on dentine discs was assessed using qPCR, as described above.

### 2.6. Statistical Analysis

All graphs, data distribution, and statistical analysis were performed using GraphPad Prism version 8 (GraphPad, San Diego, CA, USA). Before analysis, data distributions were assessed using a D’Agostino–Pearson omnibus normality test. Kruskal–Wallis with Dunn’s tests were used to determine the p values for multiple comparisons when data were not normally distributed (non-parametric data). The Mann–Whitney test was used to determine the p values for two comparisons with the non-parametric data. Differences were considered statistically significant if *p* < 0.05.

## 3. Results

### 3.1. Conception of the Endodontic Biofilm Model

First, to best determine key representative microorganisms within the endodontic biofilm model, we initially used an ontology-based analysis to identify the most frequent microorganisms present in the literature of endodontic infections over the past 30 years. After removing terms that were mentioned less than five times, a total of 222 ‘organism-related’ terms were identified. Figure 1 shows the most frequently cited microorganisms including *Streptococcus mutans* (mean = 15.3),* Candida albicans* (mean = 12.03), and *E. faecalis* (mean = 12). The frequent use of these terms suggests that these species are considered to be highly associated with endodontic infections. Moreover, while not in a statistically significant manner, the frequency of the terms *E. faecalis* and *C. albicans* showed notable decreases and increases between the years 2015 and 2020, respectively (data not shown).

To gain further insight into the present status of endodontic microbiology, we manually analyzed 21 studies that have used NGS to characterize the microbiota associated with root canal infections (Appendix A). NGS studies have shown that larger quantities of bacterial genera can be identified per canal, revealing substantially larger bacterial richness than previous non-NGS findings. Using a binary coding of present and absent to denote a role in endodontic infection, then from the 45 genera observed in these studies, at least 50% of the studies *Streptococcus*, *Prevotella, Lactobacillus*, *Porphyromonas*, and *Fusobacterium. Atopbium, Dialster*, *Pyramidobacter*, and *Actinomyces* were also observed in approximately 35% of the studies. Notably, *Enterococcus* was only observed in approximately 20% of these studies. None of these studies focused on using ITS sequences to assess the mycobiome, so no *Candida* data were available except the study by Persoon and colleagues [24]. Taken together, root canal biofilms are more complex than initially thought, where different species of *Candida*, along with a number of different acidogenic bacteria, comprise the endodontic biofilm. This was the basis for further in vitro biofilm development. 

### 3.2. Endodontic Biofilm Model Optimisation

Next, the optimal media, atmospheric conditions and incubation times were determined for the mono- and mixed-species biofilm models. For this, two supplemented media (RPMI/TSB and RPMI/THB with or without 10% fetal bovine serum (FBS)), three different atmospheric conditions (O_2_, 5% CO_2_, and AnO_2_) and two incubation times (24 h and 48 h) were selected for testing. Initial results indicated *F. nucleatum* and *P. gingivalis* developed minimal biofilm formation under any environmental condition when grown as mono-species models (Appendix A, and heatmap in Figure 2A). Conversely, *S. gordonii* formed equally dense biofilms (~1.0–1.5 A_570nm_) after 24 and 48 h in all atmospheric conditions tested (Appendix A and heatmap in Appendix A). For the *C. albicans* mono-species and mixed-species biofilms, the highest biomass was observed at both timepoints under atmospheric O_2_ and CO_2_ conditions, compared to the anaerobic environment (Appendix A and heatmap in Figure 2A). The greatest changes were significant at 24 h and 48 h for *C. albicans* only between CO_2_ and AnO_2_ conditions (Appendix A; ** *p* < 0.01 and *** *p* < 0.001, respectively). For the mixed-species biofilm, significant changes were observed at 24 h in CO_2_ conditions compared to AnO_2_ (* *p* < 0.05). However, these differences were not statistically different at 48 h (Appendix A). When comparing the media, denser biofilms were formed for *C. albicans* only when grown in TSB compared to THB. The biomass was significantly different at 24 h (*** *p* < 0.001). For S. *gordonii*, denser biofilms were formed when grown in THB, with a statistically significant difference at 24 h (Figure 2A). For the mixed-species biofilms, the biomass was not significantly different at 24 h and 48 h between the two media tested (Appendix A). Taken together, the biomass results presented here show that mixed-species biofilms grown in TSB and THB at 5% CO_2_ formed denser biofilms than those grown under normal atmospheric or anaerobic conditions. 

Next, quantitative PCR (qPCR) analyses highlighted subtle changes to the biofilm composition in the different media when the biofilms were grown at the optimal 5% CO_2_ conditions. When comparing the total colony forming equivalent (CFE)/mL of microorganisms in the mixed-species biofilms, these were comparable between the different parameters tested. For example, biofilms grown in RPMI/TSB + FBS at 24 h and 48 h were 8.83 × 10^8^ CFE/mL and 6.89 × 10^8^ CFE/mL, respectively, whereas 1.01 × 10^9^ CFE/mL and 1.30 × 10^9^ CFE/mL were the total for all microorganisms in the mixed-species biofilms grown in RPMI/THB + FBS at 24 h and 48 h, respectively (Figure 2B). With respect to % compositional changes, all biofilms were predominantly composed of *S. gordonii* and *C. albicans* (>90% and 3–9%, for both microorganisms, respectively). *F. nucleatum* and *P. gingivalis* were present in all biofilms at relatively low proportion. Total biofilm composition (%) for all microorganisms for each biofilm are shown in Table 2. These qPCR compositional analyses indicate that irrespective of inoculation media (THB vs. TSB, supplemented with FBS), the percentage composition and total CFE/mL of microorganisms in each mixed-species biofilm were comparable. 

To further optimize the model development, biofilm ultrastructure, and architecture were observed using scanning electron microscopy (SEM) (Figure 3). The four biofilms with the greatest biomass, as shown in Figure 2, were selected for imaging. All four biofilms imaged (THB/RPMI and TSB/RPMI, both with 10% FBS supplemented, grown in CO_2_ at 24 h and 48 h, respectively) were dense, with *C. albicans* yeast and hyphal cells co-aggregated with clusters of bacterial cells. These aggregates of bacterial cells were predominantly comprising of cocci-shaped colonies embedded in extracellular matrix (highlighted by white arrows in Figure 3 inset). *F. nucleatum* formed a scattered, loose filamentous network that was loosely scattered around the hyphae (highlighted by blue arrows in Figure 3 insets, at timepoint 24 h for both media). Finally, sparse colonies of *P. gingivalis* were also attached to the hyphae (highlighted in red), again mostly visible at 24 h in both media. Combined with the results from Figure 2, the ultrastructure of the biofilm was relatively comparable when grown in the two supplemented media tested, with all microorganisms particularly visible at 24 h.

### 3.3. Assessing Antimicrobials within the Optimized Endodontic Biofilm Model

We next wanted to assess whether the optimized biofilm model was a suitable system to be tested with conventional and novel therapeutics. For this, given the close similarities between the biomass, composition, and architecture of the mixed-species biofilms grown in the two types of FBS-supplemented media (Figure 2 and Figure 3), THB/RPMI was selected for testing. Furthermore, for these in vitro antimicrobial testing studies, THB/RPMI media were supplemented with and without 10% FBS to ensure that FBS did not adversely affect the efficacy of the antimicrobial challenge. In an attempt to mimic endodontic treatment regimens, biofilm testing involved either direct treatment with the chosen compound of a biofilm (termed “treatment”), or mechanical disruption of the biofilm, prior to re-growth experiments in pre-treated culture plates (termed “prevention”). Three antimicrobials were selected for testing: two conventional endodontic treatments, chlorhexidine (CHX) and EDTA, and a novel therapeutic (chitosan), which has been shown to have antimicrobial properties [25,26]. Metabolic activity results indicated that CHX was effective in the treatment of the mixed-species biofilms, in both media supplemented with and without FBS, when compared to the positive control (Figure 4A,C; **** *p* < 0.0001). Interestingly, chitosan was just as effective as CHX in the treatment of the biofilm (Figure 4A,C; **** *p* < 0.0001). EDTA did not significantly reduce the metabolic activity of the biofilm following treatment. In the “prevention” studies, all three treatments were effective in reducing the metabolic activity of the re-grown biofilms (Figure 4B,D; EDTA ** *p* < 0.01). For these studies, chitosan appeared to be most effective in reducing the level of metabolic activity of the regrown biofilm, with no detectable fluorescence recorded for chitosan-treated biofilms in the presence of FBS, whilst significant reductions in metabolic activity were seen in the absence of FBS (Figure 4C,D; both **** *p* < 0.0001). 

To further investigate the effects of the therapeutics on the mixed-species biofilms in the “prevention” group, the composition of the treated and untreated biofilms were analyzed using qPCR. These compositional analyses showed subtle fluctuations in the proportion of each microorganism in the mixed-species biofilms following treatment (Figure 5). All % changes of the four microorganisms are shown in Table 3. In brief, all biofilms were still dominated by *S. gordonii* and *C. albicans*, however, the latter was most affected during re-growth following treatment with the three therapies. Total CFE counts for the treated biofilms were all reduced compared to the positive control (untreated biofilms) during the “prevention”. The greatest change was observed for chitosan-treated biofilms, which was reduced from 2.15 × 10^8^ CFE/mL in the positive control to 3.75 × 10^5^ CFE/mL (99.8%) in the chitosan treated biofilms in media supplemented with FBS (Figure 5A, far left and far right panels). A similar reduction was observed for chitosan-treated biofilms in media without FBS supplemented, with treatment reducing the microbial burden from 4.37 × 10^8^ CFE/mL in untreated to 2.74 × 10^5^ CFE/mL (99.9%) in biofilms treated with chitosan (Figure 5B, far left and far right panels).

### 3.4. Assessing the Effect of Chitosan on the Optimized Endodontic Biofilm Model upon Dentine

Finally, the optimized biofilm model was grown on a biologically accurate substrate (dentine) to further mimic interactions between microorganisms and the host in endodontic infections. A similar experimental setup was used as described for the “prevention” studies in Figure 4 and Figure 5. Since chitosan appeared as the most effective therapeutic in preventing re-growth of the mixed-species biofilms, this was selected for testing in the biological system. Results indicated that following treatment with chitosan, there were significant reductions in the total CFE/mL of the mixed-species biofilms (Figure 6A; **** *p* < 0.0001), which was in agreement with the results described in Figure 5. The compositional changes of the biofilm (Figure 6B,C, and Table 4) indicated that the treatment reduced the levels of *C. albicans* from ~14.51% to ~3.48%, which is in line with previous publications that chitosan possesses antifungal activity [27,28,29,30,31]. Results from this and preceding sections indicate that chitosan may be an effective alternative to conventional endodontic therapeutics.

## 4. Discussion

Within the field of endodontics, there has been a clear direction of travel toward model systems with more challenging complexity, both biologically and anatomically, to more closely mirror the clinical reality. Microbiological research fits within this ethos, with clear pointers from the literature highlighting inadequacies in biofilm models used for the development of better chemotherapeutic approaches. This is best illustrated in the quote from a recent systematic review that stated, “Because of substantial variation in experimental parameters, it is difficult to compare results between studies. This demonstrates the need for a more standardized approach and a validated endodontic biofilm model” [1]. To this end, we set out to tackle this research need with the aim of creating and testing an optimized biofilm model that was both simple and reproducible, and importantly, being truly representative of endodontic infection. Using a combination of ontological analyses and biofilm optimization studies, we were able to develop and characterize a robust 4-species interkingdom model that ultimately revealed how chitosan has potential for the management of endodontic infections on dentine substrates. 

Variation in the endodontic microbiome has been shown depending on the sampling site within the canal, with unique bacterial profiles present at the apical and coronal sites of the root [32,33]. Therefore, the purpose of this study was to optimize development of a mixed-species biofilm model, guided by an ontological analysis, that contained microorganisms most closely associated with root canal infections. Furthermore, this model would be used as a testbed for assessing efficacy of conventional and novel endodontic therapeutics in vitro. Whilst we appreciate the caveats on what represents the ideal and most representative model, there was clearly a need for a model with microbial complexity. We were able to use ontological analysis and manual interrogation of the available NGS studies to show that several key species were common in these infections. Notably, *E. faecalis* does not appear at all relevant for these studies and this ought to be considered when developing assays for studying endodontic disease. Therefore, for the purpose of our study, we selected a panel of four organisms that represented a diverse and relevant population of bacteria and yeast, and with different atmospheric and nutritional requirements. We purposely limited our model to four species, which reduces complexity, supports reproducibility, but importantly, limits costs for quantitative and compositional analysis. *P. gingivalis* was preferentially selected over a *Prevotella* species due to its importance in oral health, and therefore its overall amenability for mechanistic studies [34]. We spent time optimizing the biofilm model using a range of experimental parameters such as different atmospheric conditions, media, and incubation times. Each of these parameters were assessed to ensure optimal conditions were used to support the growth of both *C. albicans* and the three bacterial species. The media selected contained a 1:1 ratio of RPMI and THB as previously described [19] or TSB, and THB was supplemented with hemin and menadione. We deemed it pertinent to investigate different types of media for our model given that others have highlighted that variation in media constituents can impact fungi–bacteria biofilm growth in vitro [19,35,36]. Therefore, RPMI was included to permit hyphal formation by *C. albicans*, whilst the supplemented bacteriological media provided important nutrients to support the growth of both *S. gordonii* (TSB/THB) and the two anaerobes, *F. nucleatum* and *P. gingivalis* (hemin/menadione supplement). Furthermore, FBS was also incorporated into the media to assess whether a rich variety of proteins such as albumin present in the FBS could aid *Candida* hyphal formation and bacterial growth in the biofilm model. Indeed, FBS supplemented media provided an increase in biomass formation in the mixed-species biofilms (Figure 2). Nevertheless, regardless of the media used, all biofilms possessed similar ultrastructure and architecture (Figure 3), whilst compositional analyses showed that the biofilms were predominated by *C. albicans* and *S. gordonii*, making up ~99% of the final composition, with the two anaerobes comprising of less than 1%. This relatively low number of *F. nucleatum* and *P. gingivalis* may be explained by the atmospheric conditions used (e.g., 5% CO_2_), which was selected as this condition gave rise to mixed-species biofilms with the greatest biomass (Figure 2). Being obligate anaerobes, these two microorganisms cannot survive in oxygenated microenvironments unless grown in the presence of oxygen-consuming species or in other oxygen-limiting conditions [37,38,39,40]. Therefore, initially, we postulated that as *C. albicans* provides a hypoxic niche for anaerobic bacteria to survive [41], allowing the anaerobic microorganisms to be more represented in the final composition. However, as the four microorganisms were added together, instead of sequentially as in our previous oral biofilm models [21,42,43,44], the microenvironment itself may not have been suitably anoxic during the first few hours to allow survival of large quantities of *F. nucleatum* and *P. gingivalis*. Nevertheless, we deemed the biofilm to be an accurate model given that, in vivo, microorganisms such as *P. gingivalis* are found in very low quantities in disease [5,45]. Importantly, all four microorganisms could be detected and observed within the biofilm.

In endodontic infections, the main objective is to eliminate the microbial burden. The optimized biofilm model used here provided a useful testbed for assessing the therapeutic effects of two conventional endodontic treatments, CHX and EDTA, and a novel antimicrobial compound called chitosan. For these in vitro studies, a “treatment” and “prevention” group were assessed, with the latter providing an in vitro imitation for root debridement, then irrigant application to prevent biofilm regrowth. From these results, it was clear that all treatments tested were effective in reducing the microbial load and/or metabolic activity of the regrown biofilm in the “prevention” group, whilst EDTA was ineffective in the “treatment” group (Figure 4 and Figure 5). We deemed it pertinent to include this “prevention” group to recapitulate an endodontist’s clinical treatment plan of an infected root canal. Indeed, we and others have previously shown with other oral biofilm models that these platforms are useful for more accurately mimicking treatment regimes in vitro [21,46]. For example, Sherry and colleagues (2016) showed that combinational treatment of brushing and denture cleansing of a 11-species denture biofilm model was more effective in reducing microbial viability than singular therapies alone. To further justify our model, others have shown that endodontic treatments such as CHX, sodium hypochlorite, and EDTA were only effective against mono-species biofilms when coupled with mechanical agitation [47,48], and that these treatment regimens are unable to fully reduce and limit biofilm regrowth of single- and dual-species biofilms [49,50].

Interestingly, the novel antimicrobial chitosan was as effective against the multi-species biofilms as the two conventional treatments (Figure 5 and Figure 6). Chitosan (poly-(β-1→4)-2-amino-2-deoxy-d-glucopyranose) is a nontoxic, naturally occurring polysaccharide produced by deacetylation of chitin, derived from the exoskeletons of arthropods and cell walls of fungi, with wide-spectrum antimicrobial activity [26]. The compound is known to possess antimicrobial activity against a variety of bacteria (both Gram-positive and Gram-negative) and fungi [30,51]. Further to this, recently our group has shown that chitosan is effective against the nosocomial pathogens *C. albicans* and *C. auris*, both in vitro and in vivo [31,52]. The potential use of chitosan in endodontic treatment is not unheard of, with a recent publication highlighting that chitosan nanoparticles, when incorporated into a paste with propolis, an extract from honey, was effective in reducing the colony forming units of an *E. faecalis* biofilm grown on human root canal dentin [53]. Another study also showed that solubilized chitosan was effective in reducing the viability of three single-species biofilms containing *Streptococcus mutans*, *Actinomyces naeslundii*, and *Enterococcus faecalis* [54]. Our group has recently shown chitosan within a nanocarrier system to be effective against interkingdom biofilms representing caries (five species), gingivitis (seven species), and denture stomatitis (11 species) [55]. Here, we show with our optimized endodontic model that chitosan is particularly effective in inhibiting regrowth of a mixed-species biofilm model following physical debridement. This was also reaffirmed on a biological substrate, bovine dentine, to further validate our model and antimicrobial results for chitosan (Figure 6). Bovine dentine is very similar to human dentine with regard to morphology, physical properties, and chemical composition [56]. Interestingly, others have shown that chitosan is able to deposit on the surface of bovine dentine slabs and coat the periphery of the dentinal tubules, which likely prevents recolonization of the biofilm [57]. However, due to the cost implications attached to the bovine enamel, we were restricted to selecting one therapeutic arm to test in this biological system. Nonetheless, results indicate that the biofilm composition is similar to that of those grown in microtiter plates or on plastic coverslips. Future studies merit consideration of repeating this biological system with different therapeutics, especially given that compounds such as CHX, EDTA, and chitosan can interfere with dentine mineralization, collagen fibers, and dentine bond strength with dental composites [54,58,59]. 

## 5. Conclusions

Interkingdom biofilms are important clinical entities, and this is readily reflected through ontological analysis of the trends developed in the literature. We have shown that a simple representative, reproducible, and effective four species endodontic biofilm model has been developed and applied to testing of known and novel endodontic irrigants. Notably, the antifungal chitosan was highly effective against the interkingdom biofilm. Given the importance of *C. albicans* in endodontic infections, then consideration of *C. albicans* as one of the ‘keystone’ structural components of interkingdom biofilms makes it a critical target for new chemotherapeutics [60]. If this can be controlled, then the co-adherent bacteria species within the biofilm can be sensitized to endodontic irrigants, which will ultimately drive better oral health outcomes through resolution of endodontic infection.

## Figures and Tables

**Figure 1 microorganisms-08-01988-f001:**
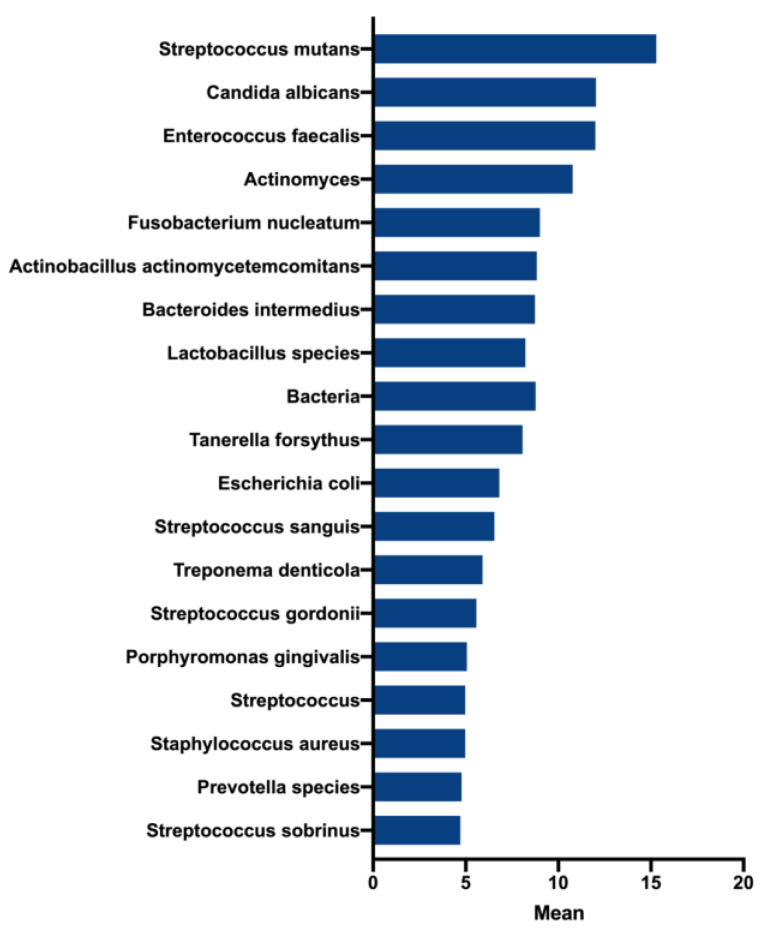
Most frequent terms and their patterns in the literature of endodontic microbiology between 1990 and 2020. The top 20 frequent organisms mentioned in the literature.

**Figure 2 microorganisms-08-01988-f002:**
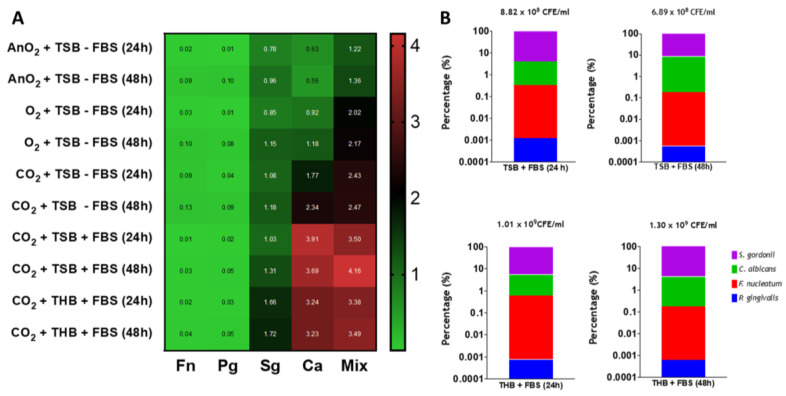
Biomass and compositional analysis of mono- and multi- species biofilms. (**A**) Heatmap analyses. Biomass quantification via crystal violet assessment of mono- and multi-species biofilms grown in three different incubation conditions (aerobically, 5% CO_2_ and anaerobically) in a 1:1 mixture of either RPMI/TSB or RPMI/THB supplemented with 10% FBS for 24 h and 48 h in 24-well flat-bottom plates. Values in heatmap show the mean absorbance reading at 570 nm for all biofilms tested; (**B**) Compositional analysis of mixed biofilms. The graphs show percentage composition using colony forming equivalents (CFE)/mL of each microorganism in the multispecies biofilms grown in a 1:1 mixture of either RPMI/TSB or RPMI/THB supplemented with 10% FBS, for 24 h and 48 h in 6-well flat-bottom plates, in 5% CO_2_. Total CFE/mL for all four microorganisms are shown above each bar graph. All experiment are representative of *n* = 3 from three independent repeats.

**Figure 3 microorganisms-08-01988-f003:**
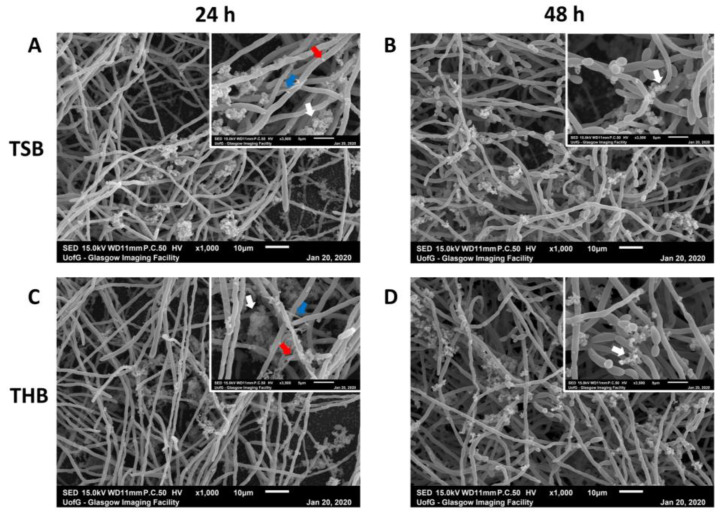
Scanning electron microscopy images of multi-species biofilms. Mixed biofilms were grown in 1:1 1640/TSB (**A**,**B**) or in 1:1 RPMI/THB (**C**,**D**) + 10% FBS for 24 h (**A**,**C**) and 48 h (**B**,**D**), before being processing for scanning electron microscopy (SEM) imaging. SEM images show rich hyphal *C. albicans* growth with clusters of bacteria adhering the hyphae. These clusters of bacteria predominantly appear characteristic of morphological streptococci (white arrows). Blue and red arrows indicate *F. nucleatum* and *P. gingivalis* shaped colonies attached to the *Candida* hyphal network. Scale bar represents 10 µm and 5 µm at ×1000 and ×3500 magnification, respectively.

**Figure 4 microorganisms-08-01988-f004:**
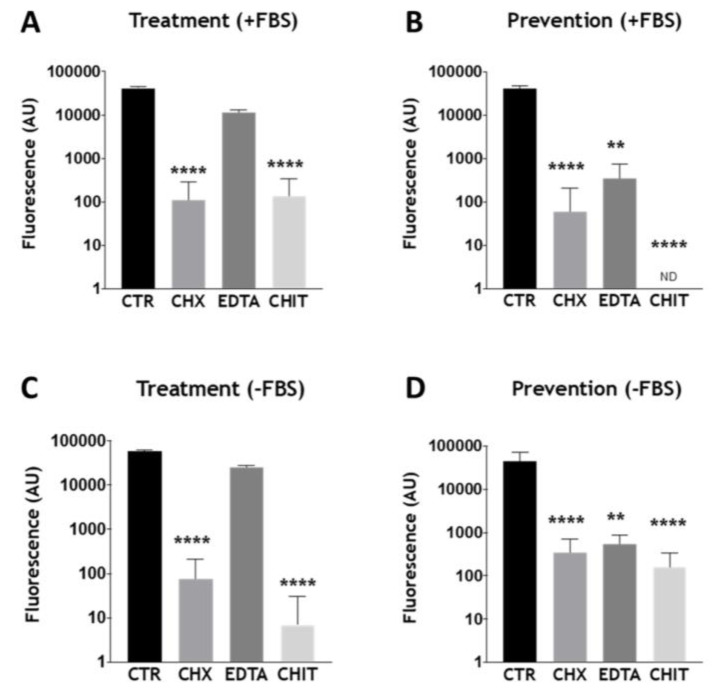
Assessing biofilm viability following treatment with conventional and novel endodontic therapies. Biofilms were grown in RPMI/THB + (**A**,**B**) or − (**C**,**D**) 10% FBS in 24-well plates for 24 h. After incubation, biofilms grown in the “treatment” group were treated with chlorhexidine (0.2%), chitosan (0.7 mg/mL), and EDTA (230 mM) (**A**,**C**). Positive controls received no treatment. Biofilms in the “prevention” group were disrupted first by scraping into RPMI/THB +/− 10% FBS media, then diluted 1:5 with fresh media (**B**,**D**). The diluted samples were then inoculated into a new 24-well tissue culture plate containing each antimicrobial test agent adjusted to twice the desired above concentrations. Plates were then incubated for an additional 24 h to permit biofilm re-growth. Positive controls were disrupted and diluted in a similar manner, however, were untreated following transfer to a fresh plate. The metabolic activity was measured by the Alamar Blue assay. Data were analyzed by Kruskal–Wallis with Dunn’s tests. The mean of each column was compared with the mean of the control group. * indicates statistically significant differences (** *p* < 0.01, **** *p* < 0.0001). Error bars represent standard deviations from a total of *n* = 3 biofilms taken from three independent experiments.

**Figure 5 microorganisms-08-01988-f005:**
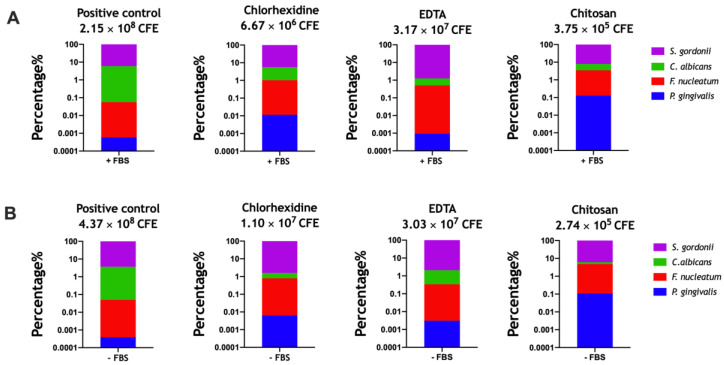
Compositional analyses of re-grown multi-species biofilms in the presence (**A**) and absence (**B**) of FCS. Biofilms in the “prevention” group as described in Figure 4 were processed for DNA extraction prior to composition analyses via quantitative polymerase chain reaction (qPCR). In brief, biofilms were disrupted, diluted 1:5, then transferred to fresh 6-well culture plates containing chlorhexidine (0.2%), chitosan (0.7 mg/mL), and EDTA (230 mM). Plates were cultured for 24 h to permit biofilm re-growth before DNA was extracted using the DNeasy Extraction Kit, following the manufacturer’s instructions. Average % composition is shown in the bar graphs, data representative of biofilms from three independent repeats (three technical replicates in each experiment).

**Figure 6 microorganisms-08-01988-f006:**
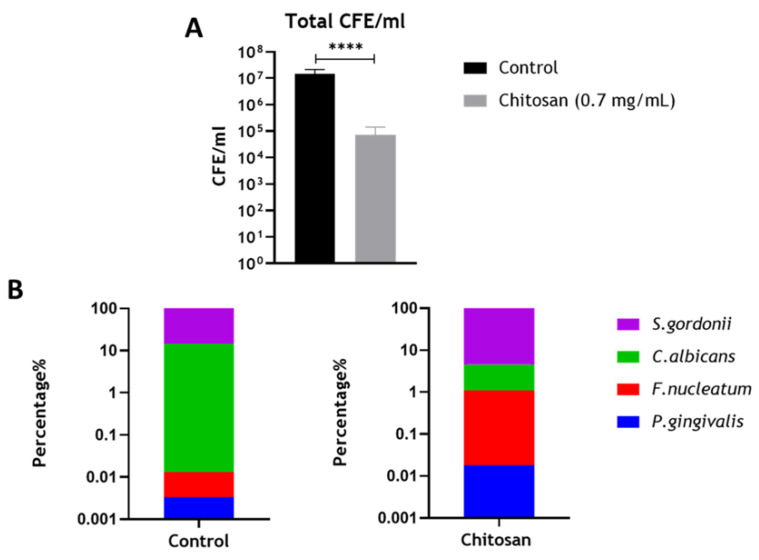
Compositional analysis of re-grown multi-species biofilm on bovine dentine discs. Biofilms were grown in RPMI/THB in 24-well plates for 24 h. After incubation, the biofilms were disrupted and diluted to 1:5 and then inoculated on pre-sterilized bovine dentine discs placed in a 24-well plate treated with chitosan at 0.7 mg/mL. Following additional 24 h incubation, the total CFE/mL (**A**) and composition of the regrown biofilms (**B**) were assessed using qPCR. Data were analyzed by the Mann–Whitney test. Data representative of *n* = 3 (**** *p* < 0.0001).

**Table 1 microorganisms-08-01988-t001:** Primer sequences used in this study for compositional analysis of multi-species biofilm models.

Organism	Forward Primer 5’-3’	Reverse Primer 5’-3’
*S. gordonii*	GATACATAGCCGACCTGAG	TCCATTGCCGAAGATTCC
*F. nucleatum*	GGATTTATTGGGCGTAAAGC	GGCATTCCTACAAATATCTACGAA
*P. gingivalis*	GGAAGAGAAGACCGTAGCACAAGGA	GAGTAGGCGAAACGTCCATCAGGTC
*C. albicans*	CTCGTAGTTGAACCTTGGGC	GGCCTGCTTTGAACACTCTA

**Table 2 microorganisms-08-01988-t002:** Percentage composition of the mixed-species biofilm model.

	Percentage Composition (%) *			
	*S. gordonii*	*C. albicans*	*F. nucleatum*	*P. gingivalis*
TSB (24 h)	95.79	3.86	0.34	0.001
TSB (48 h)	91.23	8.57	0.19	0.001
THB (24 h)	94.37	5.00	0.63	0.001
TSB (48 h)	95.82	4.00	0.18	0.001

* Average percentage composition of each of the four microorganisms in the mixed-species biofilm model grown in two different media, TSB/RPMI and THB/RPMI, both supplemented with FBS. Biofilm composition was assessed at 24 h and 48 h. Values representative of *n* = 3 from three independent experiments.

**Table 3 microorganisms-08-01988-t003:** Percentage composition of the mixed-species biofilm model following “prevention” treatment.

	Percentage Composition (%) *							
	*S. gordonii*		*C. albicans*		*F. nucleatum*		*P. gingivalis (%)*	
	−FBS	+FBS	−FBS	+FBS	−FBS	+FBS	−FBS	+FBS
+CONT	96.30	93.98	3.65	5.96	0.05	0.06	0.0004	0.001
CHX	98.41	94.36	0.78	4.61	0.79	1.02	0.006	0.011
EDTA	97.91	98.73	1.74	0.76	0.38	0.51	0.003	0.001
CHITOSAN	93.85	92.10	1.35	4.42	4.68	3.36	0.110	0.130

* Average percentage composition of each of the four microorganisms in the mixed-species biofilm model grown in THB/RPMI supplemented with or without FBS following treatment. Pre-grown biofilms were manually disrupted with a cell scraper, diluted 1:5 in THB/RPMI supplemented with or without FBS, then treated with chlorhexidine (CHX), EDTA, or chitosan for a further 24 h prior to biofilm compositional analysis. Positive control biofilms received no treatment during regrowth. Values representative of *n* = 3 from three independent experiments.

**Table 4 microorganisms-08-01988-t004:** Percentage composition of the mixed-species biofilm model grown on dentine discs following “prevention” treatment.

	Percentage Composition (%) *			
	*S. gordonii*	*C. albicans*	*F. nucleatum*	*P. gingivalis*
+ CONT	85.54	14.51	0.01	0.003
CHITOSAN	95.42	3.48	1.09	0.018

* Average percentage composition of each of the four microorganisms in the mixed-species biofilm model grown in THB/RPMI. Pre-grown biofilms were manually disrupted, diluted 1:5 in THB/RPMI, then treated with chitosan for a further 24 h prior to biofilm compositional analysis. Positive control biofilms received no treatment during regrowth on the biological substrates. Values representative of *n* = 3.

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
