# Peer review of "Filling the Void: An Optimized Polymicrobial Interkingdom Biofilm Model for Assessing Novel Antimicrobial Agents in Endodontic Infection"

_microorganisms, 2020, doi:10.3390/microorganisms8121988_

Round 1
Reviewer 1 Report
What is the green swatch on line 196.
The concept of a "keystone" organism is approrpiate, nonetheless, a cadre of sympathetic organisms may in fact be the "keystone organism group" culprits in these infections.
Author Response
We thank the review for reading our article and providing positive feedback.
"What is the green swatch on line 196." - apologies, this was an edit from an author we did not finalise. Its been sorted now to show a 2 min denaturing cycle.
"The concept of a "keystone" organism is approrpiate, nonetheless, a cadre of sympathetic organisms may in fact be the "keystone organism group" culprits in these infections."
This is an interesting point. It is not possible to state unequivocally that one organism is "the" keystone, a problem that has dogged the focus of P. gingivalis for years. However, as a mycologist, it is clear that Candida has been ignored and underappreciated. This paper was submitted with a companion paper supporting our studies into the importance of Candida in complex communities. We have softened our final conclusions on line 559 to this effect "C. albicans as one of the ‘keystone’ structural components". We hope this echoes your statement.
Reviewer 2 Report
The author optimized the condition for polymicrobial interkingdom biofilm formation and analyzed the antimicrobial activity of CHX, EDTA and chitosan against optimized endodontic biofilm model. The manuscript is clearly written, the methodology is appropriate, and the results are well presented. There is only one issue concerns me. Is there any rationale for the author to choose the combination of the strains to form the biofilms? For example, although S.gordonii is on the top 20 list, S.mutans is the top 1, why the author picked S.gordonii rather than S.mutans?
Author Response
We thank the reviewer for their positive feedback and queries.
"There is only one issue concerns me. Is there any rationale for the author to choose the combination of the strains to form the biofilms? For example, although S.gordonii is on the top 20 list, S.mutans is the top 1, why the author picked S.gordonii rather than S.mutans?"
This is an interesting question. The reality is that we don't really know which streps are the most important. Historically, culture techniques make it difficult to differentiate these different streps, and using NGS techniques the resolution for differentiating streps is simply not available using illumina platforms. For the purposes of this study we erred on the side of caution and selected a strep species that we know positively interacts with Candida, as this has greater potential for modelling interkingdom interactions at a mechanistic level in addition to antimicrobial studies. Its a good idea to evaluate S. mutans going forward to do a side by side comparison, and we are already planning this.
Best wishes,
Gordon